# Comparative Assessment of Oxidative and Antioxidant Parameters in Mule and Horse Neonates during Their First Month of Extrauterine Adaptation

**DOI:** 10.3390/ani13243878

**Published:** 2023-12-16

**Authors:** Amanda Vallone Riccio, Barbara Kolecha Costa, Maria Augusta Alonso, Fernanda Jordão Affonso, Danilo Souza França, Marcilio Nichi, Carla Bargi Belli, Amy Katherine McLean, Yatta Linhares Boakari, Claudia Barbosa Fernandes

**Affiliations:** 1Department of Animal Reproduction, School of Veterinary Medicine and Animal Science, University of Sao Paulo, Sao Paulo 05508-270, Brazil; amandariccio@gmail.com (A.V.R.); barbarakolecha@outlook.com (B.K.C.); gutalonso@gmail.com (M.A.A.); fernandajordao.vet@gmail.com (F.J.A.); danilofranca26@gmail.com (D.S.F.); mnichi@usp.br (M.N.); 2Department of Internal Medicine, School of Veterinary Medicine and Animal Science, University of Sao Paulo, Sao Paulo 05508-270, Brazil; cbbelli@usp.br; 3Department of Animal Science, University of California Davis, Davis, CA 95616, USA; acmclean@ucdavis.edu; 4Department of Large Animal Clinical Sciences, College of Veterinary Medicine and Biomedical Sciences, Texas A&M University, College Station, TX 77843, USA

**Keywords:** mules, equids, neonates, lipid peroxidation, oxidative stress, thiobarbituric acid, bilirubin

## Abstract

**Simple Summary:**

Numerous postnatal transformations occur because of physiological adjustments to oxidative circumstances, which induce reactive oxygen species formation. We studied two cohorts of 11 neonatal mules and 11 neonatal horses to assess oxidative and antioxidant profiles during the period of extrauterine adaptation. The neonatal mules exhibited lower levels of lipid peroxidation along with an elevated concentration of glutathione peroxidase. Furthermore, bilirubin concentrations were notably reduced in the neonatal mules. Consequently, our findings lead us to infer that neonatal mules manifest distinct oxidative activities and antioxidant capabilities in comparison to their equine counterparts.

**Abstract:**

After parturition, a rapid transition occurs from the intrauterine to the extrauterine milieu, exposing neonates to physiological circumstances characterized by oxidative conditions that instigate the generation of reactive oxygen species. These free radicals play pivotal roles in physiological processes; however, an imbalance between their production and the removal of antioxidants can result in severe cellular damage. The main objective of this study was to compare the oxidative and antioxidant profiles in mule and horse neonates immediately post-parturition, as well as at subsequent time points (1, 6, 12, and 24 h, 7 and 30 days) during their extrauterine existence. The parameters assessed included the systemic concentrations of Thiobarbituric Acid Reactive Substances (TBARS) and carbonyl groups; the activities of the antioxidants superoxide dismutase (SOD) and glutathione peroxidase (GPx); and the levels of the total, indirect, and direct bilirubin. Our results showed no interaction effect between the neonatal groups and the assessed time points for the variables under investigation. Notably, the concentrations of TBARS, as a marker of lipid peroxidation, and bilirubin were consistently lower in the mules, whereas the glutathione peroxidase (GPx) activity exhibited higher levels in this group. The bilirubin levels were notably reduced in the mule neonates. The TBARS demonstrated a progressive decrease over the observation period in both groups, while the GPx activity remained relatively stable from birth to 7 days, with a substantial increase evident at the 30-day mark. Protein oxidation was not affected by the group and time, while for the SOD values, all times were statistically similar, except for the lower activity at T1h. Consequently, our findings lead us to the conclusion that neonatal mules and horses manifest distinct patterns of oxidative activity and antioxidant capacity during the initial month of their extrauterine existence, potentially indicative of different adaptation mechanisms to the extrauterine environment.

## 1. Introduction

Newborns must undergo physiological adaptations under the new pro-oxidative environment of high levels of oxygen and the consequent increase in reactive oxygen species (ROS) [1]. The most important ROS are the superoxide anion (·O_2_^−^), hydroxyl radical (·OH), and hydrogen peroxide (H_2_O_2_) [2]. These free radicals are important during this adaptation phase, but an excess or imbalance of the antioxidant mechanisms can trigger oxidative stress, which causes biomolecule, protein, lipid, carbohydrate, and DNA structural damage [3,4].

The defense against free radicals is divided into non-enzymatic and enzymatic antioxidants. The latter is composed of the enzymes superoxide dismutase (SOD), glutathione peroxidase (GPx), catalase, and peroxiredoxin. While SOD catalyzes the dismutation of the superoxide radical (·O_2_^−^), converting it into oxygen and hydrogen peroxide (H_2_O_2_), GPx, and catalase prevents the accumulation of hydrogen peroxide. An imbalance of this system may induce ·OH production, which has the greatest reactive potential, causing oxidative damage [5,6]. On the other hand, the non-enzymatic defense mechanism is composed of reducers of free radicals. Among low-molecular-weight biological non-enzymatic antioxidants, bilirubin stands out [6,7]. Currently, it is known that under physiological conditions, bilirubin has a potent antioxidant role [8] and is highly lipophilic, protecting against oxidative lipid degradation.

The antioxidant and oxidative states have been studied in different domestic animal species, with emphasis on the gestational, parturition, and neonatal periods in equine [7,9], bovine [8], ovine [10,11], and porcine [12] species. However, there are no published reports of oxidative and antioxidant profile in mule foals. Nevertheless, these biomarkers are important as they are one of the central elements in signal transduction pathways involved in cell proliferation, differentiation, and apoptosis [13]. A recent study [14] found physiological differences between horse and mule neonates, such as a lower concentration of the steroid 3β,20α-dihydroxy-DHP, which may be related to the placental steroidogenic capacity during hybrid gestation. A higher APGAR value was also observed in mules when compared to horse foals [14], probably due to the release of progestogens in the perinatal period. These differences suggest that mule foals’ metabolism is derived from different origins, presumably placenta, gonads, and adrenals, indicating that the endocrinology of mules is different and perhaps that they undergo distinct aspects of adaptation during the first days of life. Thus, the objective of the present study was to evaluate the oxidative profile by measuring the GPx, SOD, bilirubin, TBARS, and protein oxidation activity in healthy mule neonates, and to compare it with healthy horse neonates at birth, 1, 6, 12, and 24 h, 7 and 30 days after birth.

## 2. Materials and Methods

Twenty-two neonates consisting of 11 mules originating from Pêga donkeys and Mangalarga Marchador Paulista mares, 6 females and 5 males; and 10 Mangalarga Paulista horses foals, 5 males and 5 females, were evaluated during two seasons (2015/2016 and 2016/2017) at Fazenda Santa Rita II, Piracaia, São Paulo, Brazil (latitude 23°03′14″ S and longitude 46°21′29″). Pregnancies were obtained with natural mounts, as described previously, by healthy mares [15,16]. All parturitions were monitored and the inclusion criteria were as follows: (1) neonates born from full-term eutocic parturition; (2) APGAR score ≥ 7 at birth [17]; (3) IgG ≥ 800 mg/dL at 24 h of life (IgG Check—Test for IgG detection in foals, Vencofarma^®^, Dechra, Londrina, Brazil); and (4) healthy neonates at the time of sampling (normal physical, hematological, and biochemical exams). The experiment was approved and developed according to the guidelines of the Ethics Committee on Animal Use of the School of Veterinary Medicine and Animal Science of the University of Sao Paulo, under protocol number 6001260715.

Blood was collected by jugular venipuncture in sterile vacuum tubes containing EDTA (BD Vacutainer^®^, BD, Brazil) and dried with blasted clot activator on the wall to collect serum (BD Vacutainer^®^, BD, Brazil) at birth (immediately after foal expulsion; T0h), 1 h (T1h), 6 h (T6h), 12 h (T12h), 24 h (T24h), 7 days (T7d), and 30 days (T30d) after birth. After collection, the tubes were centrifuged (150× *g* for 10 min) to separate plasma and serum. Serum (300 μL) was aliquoted into 1.5 mL amber cryotubes for total and direct bilirubin analysis. Also, 100 μL of plasma was placed into 1.5 mL cryotubes for analysis of carbonyl groups and 300 μL of plasma for TBARS testing. All samples were stored at −20 °C until analyses were completed in the Laboratory of Andrology of the Department of Animal Reproduction (School of Veterinary Medicine and Animal Science, University of São Paulo, São Paulo, Brazil). Bilirubin was analyzed in the Clinical Laboratory of the Department of Internal Medicine in the same institution.

After plasma samples were thawed, lipid peroxidation was assessed by measuring the Thiobarbituric Acid Reactive Substances (TBARS). These evaluations were made according to the protocol described previously [17]. The method is based on the reaction of two molecules of thiobarbituric acid with one molecule of malondialdehyde (MDA) at high temperatures and low pH, resulting in a pink chromogen that can be quantified with a spectrophotometer, with a wavelength of 532 nm (Ultrospec 3300 pro^®^, Amersham Biosciences, Slough, UK).

Protein oxidation was assessed by measuring carbonyl group level using a methodology based on Odetti et al., 1996 and Levine et al., 1990 [18,19], using the protocol modified and standardized for blood plasma in equine species [20]. Oxidation measurement was performed by the reaction between 2,4-dinitrophenylhydrazine (DNPH) and the carbonyl groups. Upon reaction, there was Schiff base formation to produce hydrazone, which was quantified by spectrophotometry (Ultrospec 3300 pro^®^, Amersham Biosciences, UK).

Quantification of GPx activity in blood plasma of neonates followed the technique described by NICHI et al., 2006 [14,21]. This method is based on the measurement of NADPH consumption. The reaction between a hydroperoxide and reduced glutathione (GSH) is induced and is catalyzed by GSPH-Px together with the enzyme glutathione reductase (GSSGr). This reaction causes the conversion of glutathione disulfide (GSSH—oxidized glutathione) to GSH, which in turn consumes NADPH, measured in a spectrophotometer (Ultrospec 3300 pro^®^, Amersham Biosciences, UK).

The enzymatic activity of SOD was measured indirectly, through the reduction of cytochrome C by superoxide anion (·O_2_^−^). The xanthine-xanthine oxidase system continuously generated ·O_2_^−^ (which reduced cytochrome C). SOD present in the sample competed with cytochrome C by converting the superoxide free radical to H_2_O_2_ and ·O_2_^−^, thus decreasing the rate of cytochrome C reduction, according to a technique described previously [22]. During the assay, absorbance was determined every 5 min in a spectrophotometer (Ultrospec 3300 pro^®^, Amersham Biosciences) equipped with a temperature regulator maintained at 25 °C.

Serum aliquots were analyzed with a Randox^®^ reagent kit for total bilirubin (TB) (BR 3859) and direct bilirubin (DB) (BR 3807) using the ion-selective electrode method in a Randox^®^ (Daytona model, Milton Keynes, Buckinghamshire, UK) automatic biochemical analyzer. Indirect bilirubin (IB) was obtained by the difference between total and direct bilirubin.

### Statistical Analysis

Statistical analysis was performed using the software SAS System for Windows (SAS^®^ version 9.3, SAS Institute Inc., Cary, NC, USA). Data were previously tested for normality of residuals (Gauss distribution) and homogeneity of variances by the Guided Data Analysis application. Transformations (logarithm in base 10, square root, square of the values) were used whenever necessary. Variables were analyzed by the PROC MIXED using a linear mixed model for repeated measures over time, with group and age as fixed effects. Interaction between factors was evaluated. The significance level used to reject the H0 (null hypothesis) for all variables was 5%. Results are presented as means (mean ± standard error of the mean) of the original data. Variables that presented significant effects were subjected to Tukey’s test.

## 3. Results

### 3.1. Oxidative Markers

There was no interaction between time and group for the variables TBARS (*p* = 0.48) and protein oxidation (*p* = 0.21) (Table 1). We observed no effect of the group on the protein oxidation (*p* = 0.37), while the TBARS values showed a difference between the groups, being lower in the mule neonates (*p* = 0.01) (Table 1). Lipid peroxidation, evaluated by the TBARS measurement, was affected by time (*p* < 0.0001), showing a progressive decrease over time (T0h = 159.13 ± 16.01; T30d = 88.06 ± 6.82) (Figure 1), while carbonyl groups, evaluated by protein oxidation, was not effected by time (*p* = 0.07). As there was no effect of the group, the results for the mule and horse foals are presented together.

### 3.2. Antioxidant Markers

#### 3.2.1. Superoxide Dismutase (SOD) and Glutathione Peroxidase (GPx)

For the variables SOD and GPx, there was no interaction between time and group (SOD, *p* = 0.50; GPx, *p* = 0.36; Table 2). The effect of the group was observed for GPx (*p* = 0.01) but not for SOD activity (*p* = 0.18) (Table 2). The effect of time was observed for both SOD (*p* = 0.005) and for GPx (*p* = 0.0001). Regarding the SOD values, all times were statistically similar, except for T1h, which showed a lower activity compared to T6h, T7d, and T30d (*p* = 0.005). In contrast, for GPx, increased activity was observed at T30d (*p* < 0.0001) (Figure 2), with no difference between the groups; thus, the data are shown pooled.

#### 3.2.2. Bilirubin

Regarding bilirubin, there was no interaction between time and group (TB, *p* = 0.47; DB, *p* = 0.97; IB, *p* = 0.32; Table 3). We observed that all three variables, TB, DB, and IB, were higher in the horse foals (*p* < 0001), as shown in Table 3.

There was an effect of time for all three variables (TB, *p* < 0.0001; IB, *p* < 0.0001, and DB, *p* = 0.004). The total bilirubin was statistically higher at 12 h (T12h = 2.51 ± 0.17) and 24 h (T24h = 2.96 ± 0.28), differing between 0 h, 1 h, 7 d, and 30 d (Figure 3). Notably, the TB was lowest at 0 h, 1 h, 7 d, and 30 d. The changes in the total bilirubin are directly related to the indirect bilirubin, with the indirect bilirubin showing a similar trend to the direct bilirubin (Figure 3). The direct bilirubin concentrations had a slight decrease after birth; however, the concentrations returned to similar levels from the immediate time after birth (Figure 3). Data for groups are shown combined.

## 4. Discussion

To the best of our knowledge, this study represents a novel investigation into the assessment of the oxidative status in mule neonates, while concurrently comparing them to their horse counterparts. Our evaluations showed certain distinctions in the oxidant and antioxidant profiles between these groups. Furthermore, there were also temporal variations in the parameters. Particularly, there were lower levels of lipid peroxidation and bilirubin concentration, coupled with elevated levels of glutathione peroxidase (GPx) activity observed in the mule neonates during the initial month post-foaling when compared to the horse neonates. These novel findings offer valuable insights into understanding the normal neonatal physiology and oxidative–antioxidative equilibrium specific to mules, thereby elucidating important aspects of our knowledge in this field.

Studies with mules, especially neonates, are still scarce, especially with regard to their oxidative and antioxidant profile. In humans, a reduction in lipid peroxidation, such as the adverse reactions of nitric oxide, nitrates, and nitrites, associated with an increase in GPx activity is reported [6], in agreement with our data, where mule neonates had a lower concentration of TBARS associated with a higher GPx activity. 

Our results indicate a different, but equally efficient, balance of oxidative and antioxidant values during the evaluated time points in mule neonates when compared to horse neonates, as all animals in the present study were healthy. Furthermore, when evaluating the antioxidant profile, the mule neonates showed higher GPx activity, responsible for the reduction of hydrogen peroxide, thus decreasing lipid peroxidation, which could explain the lower concentration of TBARS observed in the mule group [5]. 

The TBARS concentrations were higher at birth in all of the studied neonates, with a progressive decrease, in agreement with the data observed in equine [20] and porcine neonates [13]. Additionally, the TBARS concentrations were higher in the studied animals during the first hours of life when compared to 7 days after birth. In humans, neonates born by vaginal or cesarean section deliveries show no differences in TBARS. However, the two groups of neonates in this study had a higher concentration of TBARS soon after birth and remained high until three days of life. The oxidative capacity also remained stable during these first three days and the increased demand for oxygen due to increased metabolism may influence lipid peroxidation [21]. Additionally, we observed a progressive decrease in lipid peroxidation (TBARS) from birth to thirty days of life, which has not been reported in previous studies in newborn horses, but has also been observed in newborn dogs, with lower lipid peroxidation during the first hour of life [22]. Of note, there were no differences in the protein oxidation between the mule and horse neonates or during the evaluated time, suggesting that this parameter is consistent during the initial adaptation of the neonates to the extrauterine environment.

When we analyzed the effect of time on GPx activity, we obtained a constant concentration from birth to 7 days. However, at 30 days, there was a significant increase in GPx activity along with a lower concentration of TBARS. Previous studies in horses have demonstrated a significant decrease in TBARS concentrations from 5 min after birth to 72 and 168 h postpartum; this reduction indicates that they showed peripartum peroxidation [7]. In another study, horse neonates showed high levels of TBARS at birth, with a subsequent decrease up to 7 days, and lower GPx activity at birth with an increase at 12 h of life [6]. On the other hand, in calves, GPx activity was similar at the day of birth, pre-colostrum intake, and at 3, 7, 14, and 21 days of life [8], showing, along with previous studies in horses, a different pattern from what was observed in the present study. We found a decrease in the marker for lipid peroxidation, combined with an increase in an antioxidant enzyme, occurred at the end of the thirtieth postpartum day, indicating that there was stability in the oxidative parameters in the first month of life for these equid neonates.

There were no differences between the species in terms of the SOD activity, suggesting that the dismutation of the ·O_2_^−^ radical, which is converted into oxygen and hydrogen peroxide, was similar in both the mule and horse neonates. It was also proposed in an in vitro study with nervous system cells that bilirubin had an inhibitory activity on the superoxide radical, which would cause stability in the SOD activity. This would corroborate the data observed in this study, where we had the highest concentrations of IB at 6, 12, and 24 h of life, and the SOD activity did not vary at these moments. Similar results were found in humans [23] and equine [7] neonates, where no variations were observed in the SOD enzymatic activity, while the indirect bilirubin was elevated.

A lower concentration of total, direct, and indirect bilirubin were observed in the mules when compared to horses. In horse neonates, a physiological increase in indirect bilirubin during the first week of life has been reported, probably due to a rapid rate of the recurrence of fetal red blood cells [16,23,24]. However, studies have reported an antioxidant role of bilirubin in human serum, protecting cell membranes from lipid peroxidation due to its lipophilic properties [25]. Therefore, our hypothesis is that mule neonates may use bilirubin to minimize lipid peroxidation during the first hours of life, as the values of the total and indirect bilirubin were inversely proportional to lipid peroxidation, except for the seventh day of life.

Our data suggest different adaptation mechanisms to the extrauterine environment in mule neonates, as observed previously [16]. A higher APGAR score and different neonatal parameters were observed during the first hours of life in the mules when compared to the equine neonates. These findings suggest that mule neonates exhibit a faster adaptation to the postnatal environment. This accelerated adjustment could be attributed to their enhanced resilience and cognitive capabilities, possibly from genetic heterosis [23,26,27]. As a result, mule neonates tend to demonstrate a faster adaptation to their surroundings when compared to their horse counterparts [16,24]. 

## 5. Conclusions

Therefore, we conclude that the mules exhibited distinct oxidative and antioxidant capacity compared to the horse neonates within their initial month of life, although both mechanisms were effective. This observation may indicate distinct adaptation mechanisms to the extrauterine environment, enriching our knowledge on mule physiology.

## Figures and Tables

**Figure 1 animals-13-03878-f001:**
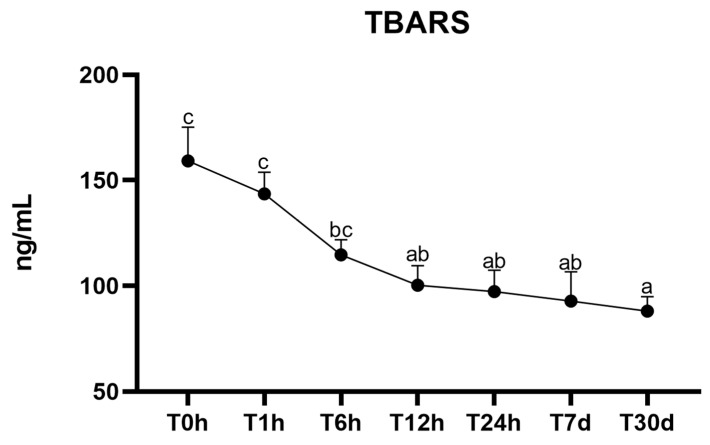
Mean ± standard error of lipid peroxidation (TBARS) in mule and horse neonates’ plasma collected from birth to 30 days after birth. ^a–c^ Different lowercase letters represent statistical difference (*p* < 0.05). Units of measurement: TBARS in ng/mL; T0h: at birth; T1h: 1 h; T6h: 6 h; T12h: 12 h; T24h: 24 h; T7d: 7 days; and T30d: 30 days.

**Figure 2 animals-13-03878-f002:**
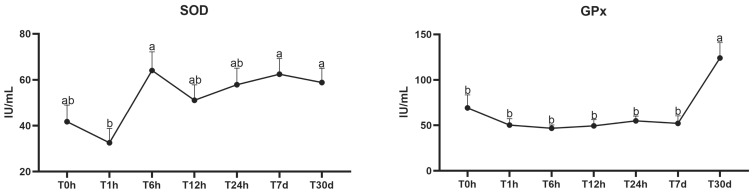
Variation in the enzymatic activities of superoxide dismutase (SOD) and glutathione peroxidase (GPx) in mule and horse neonates from birth to 30 days after birth. ^ab^ Different lowercase letters represent statistical difference for SOD and GPx. Units of measurement: SOD and GPx in IU/mL T0h: at birth; T1h: 1 h; T6h: 6 h; T12h: 12 h; T24h: 24 h; T7d: 7 days; and T30d: 30 days.

**Figure 3 animals-13-03878-f003:**
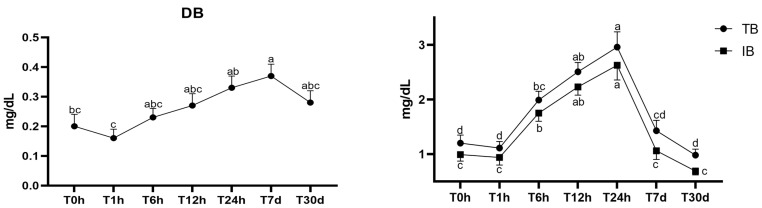
Variation in total, direct, and indirect bilirubin in mule and horse neonates from birth to 30 days after birth. ^a–d^ Different lowercase letters represent statistical difference. TB, DB, and IB units of measurement: mg/dL; TB: total bilirubin; DB: direct bilirubin; IB: indirect bilirubin T0h: at birth; T1h: 1 h; T6h: 6 h; T12h: 12 h; T24h: 24 h; T7d: 7 days; and T30d: 30 days.

**Table 1 animals-13-03878-t001:** Mean ± standard error of lipid peroxidation (TBARS) and carbonyl groups (protein oxidation) in mule and horse neonates’ plasma collected from birth to 30 days after birth.

Variables	Groups		*p*-Value	
	Mule	Horse	Group	Group × Time
TBARS	107.92 ± 6.88	118.07 ± 5.71	0.01	0.48
Protein Oxidation	2.27 ± 0.23	2.27 ± 0.30	0.37	0.21

*p* < 0.05 in the same row indicates significant difference. Units of measurement: TBARS in ng/mL; Protein oxidation in carbonyl mm/mg protein.

**Table 2 animals-13-03878-t002:** Mean ± standard error of the enzyme activity of superoxide dismutase (SOD) and glutathione peroxidase (GPx) in mule and horse neonates’ plasma collected from birth to 30 days after birth.

Variable	Group		*p* Value	
	Mule	Horse	Group	Group × Time
SOD	49.82 ± 3.93	56.38 ± 3.77	0.18	0.50
GPx	70.89 ± 5.44	57.09 ± 5.49	0.01	0.36

*p* < 0.05 in the same row indicates significant difference. Units of measurement: SOD and GPx in IU/mL.

**Table 3 animals-13-03878-t003:** Mean ± standard error of total bilirubin, direct bilirubin, and indirect bilirubin concentrations in mule and horse neonates’ serum collected from birth to 30 days after birth.

Variables	Groups		*p* Value	
	Mule	Horse	Group	Group × Time
Total bilirubin	1.31 ± 0.09	2.0 ± 0.13	<0.0001	0.47
Direct bilirubin	0.18 ± 0.01	0.36 ± 0.02	<0.0001	0.97
Indirect bilirubin	1.13 ± 0.09	1.84 ± 0.12	<0.0001	0.32

*p* < 0.05 in the same row indicates significant difference. Units of measurement: mg/dL.

## Data Availability

The data presented in this study are available on request from the corresponding author.

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
