# Peer review of "Comparative Assessment of Oxidative and Antioxidant Parameters in Mule and Horse Neonates during Their First Month of Extrauterine Adaptation"

_animals, 2023, doi:10.3390/ani13243878_

Round 1

Reviewer 1 Report

Comments and Suggestions for Authors

Dear authors,

The article is interesting and provide additional data about oxidative and antioxidants profiles in mule and horse neonates. However, there are some points that need to be very clear to the reader, and before considering the article to publication they need to be clarified.

 They are listed below:

Abstract

Line 37 – After results for TBARS and GPx, add obtained results of bilirubin and marker of protein oxidation between two species.

 Introduction

Line 73- Is 3β,20α-dihydroxy-DHP steroid hormone? I suggest that authors add explanation in front of “3β,20α-dihydroxy-DHP”.

 Lines 79-82 - The title of the paper, the obtained results and the objectives are not clearly aligned, please revise it.

 Materials and methods

 Line 84 – Please indicate sex of mule and horse.

 Have there been any premature birth? Was there an assisted delivery? Whether the neonates were conceived through artificial insemination or by natural conception. Please indicate in Materials and Methods.

 Line 101 – Be specific, indicate the parameter of protein oxidation and use further in the text and table.

 Line 139 – Add subtitle Statistical analysis

Results

Line 149 - I suggest to replace subtitle with  “3.1. Oxidant markers “

Line 155-  Although the tested parameter for protein oxidation had no effect on time, the authors need to present (in Figure 1) dynamics of this marker in equids during 30 days after birth.

Line 157 - I suggest to replace subtitle with 3.2. Antioxidant markers….which includes SOD, GPx and BIL

 Figures 1-3: Explain in the section Results why you pooled mule and horse results related dynamics of tested markers.

 Match the titles of Figures 1-3 e.g equids (Fig.1) or mule and horse (Fig.2 and 3), mean+sd or variation…

 Disscusion

The discussion needs to be restructured. Given that the author compared the oxidative stress markers between mule and horse neonates (which is the title of the paper), the focus should be on the discussion of the resulting differences in TBARS, GPx and bilirubin between two species.  

 Also, discussion related to the parameter reflecting protein oxidation is missing.

Line 205:   Oxidative stress is a condition that can be assessed by evaluating oxidants and antioxidants biomarkers. Be specific authors also evaluated antioxidants not only "oxidative status".

Lines 206-207: “Our evaluations showed distinctions in the oxidative profiles between these groups, which also exhibited temporal variations” This sentence does not correspond to the obtained results. Add "certain" distinction, because you did not showed differences in all parameters. Also, for “oxidative profiles” see comment above. “…. temporal variations” there were differences in the dynamics of pooled samples not between species. Please revise it.

 Lines 209: add compared to horse neonates

 Line 211: add oxidant-antioxidant equilibrium instead oxidant equilibrium.

 Lines 213-220: I suggest placing this paragraph near the end of the discussion, after the comments obtained results and compare to previous studies related to oxidative stress parameters.

 Line 222: indicate LP marker instead lipid peroxidation.

 Lines 225-226: The sentence is not clear enough. Does the "different" balance relate to mules versus horses. Please change “profile” into a suitable word that makes sense with “balance”.

 Lines 231-233: A sentence is too long and loses its meaning, please revise it.

Line 250: Misquoted Reference (6)? Are the authors referring to reference (8)? Check it.

Line 267: This may be one explanation for low bilirubin during the first hours of life, but for both species (Figure 3).

 How do you explain higher values of GPx and lower values of bilirubin in mules compared to horses.

 References

 list the references according to the instructions of the journal.

Comments on the Quality of English Language

No need to edit English language. Good quality. 

Reviewer 2 Report

Comments and Suggestions for Authors

This is an interesting study that adds new and valuable information about newborn mules. Some suggestions are detailed below for a better understanding of the results.

P1L6: after Barbosa Fernandes, there is an and…looks like some information is missing.

P2L52: authors could consider including some information about what or when imbalances/excess can occur in newborns.

Material and Methods

P2L84: can authors give more information about the newborn mules, for example are they also from Mangalarga Paulista mares? Was the donkey from any breed and was the same breed or type used for all mules?

Was the mare´s health status known? Also, were mares checked for their antioxidant status? For example, was GPx activity or selenium status within normal ranges? If so, please include.

P3L143-145: please use the same concept here and in the manuscript for time or age. What was the random effect? Was the time effect also included? Interaction between group x time also should be included.

Results

As a suggestion, results should be text and table/figure together, for example 3.1 lipid peroxidation and protein oxidation text (L150-156) then table 1 and figure 1. It´s easier to read and follow the results. Then, 3.2 enzymatic antioxidant profile (table 2 and figure 2), then 3.3 bilirubin (table 3, figure 3).

Figure 1 (even though no interaction was seen between time and group) must show a line for mules and one for horses. Same for figure 2 and 3.

P4L165-168: information should be in section 3.1.2 SOD and GPx.

P4L169-174: there are other statistically significant differences in TB and IB between time points, for example, T0 and T1 with T6.

Section 3.2 Figures, Tables and Schemes could be eliminated and put with the correspondent information as suggested before. Also, no schemes are presented.

Discussion

P6L221-230: some information regarding horse neonates and oxidative/antioxidant profile could be included (results were expected or within normal range?). As for humans, is also in horses a decrease in lipid peroxidation and an increase in GPx been reported?

P6L233-234: why authors think TBARS concentration was significantly higher in the first hours compared to 7 and 30 days after birth? Similar to P7L239.

Reviewer 3 Report

Comments and Suggestions for Authors

The manuscript animals-2722367 reports data on the oxidative-antioxidant status of mule and horse neonates, monitored during the first month after birth.

The research is interesting, the results are new and increase knowledge in the field. The manuscript is well-organized and well-written. Although the discussion is sufficiently argued, I suggest for it a little implementation with more comments about the obtained results on antioxidant enzymes.

I consider this paper appropriate for publication in Animals, and I suggest accepting it after the suggested revisions and with a re-reading to solve some typos in the text.

MAJOR POINTS

1) Page 2, lines 56-57. “The latter is composed of the enzymes superoxide dismutase (SOD), glutathione 56 peroxidase (GPx), and catalase.”. The authors should also include peroxiredoxin, which is a key enzyme in the antioxidant defence system.

2) It is not clear to me why the authors present a single graph that includes data from both the horse and the mule. This is also strange in light of the fact that the authors themselves state that there are differences between these two animals, which are also supported by statistics. In my opinion, the authors should produce graphs of both organisms for each parameter analysed so that the differences can be clearly seen.

3) Page 6, lines 207-210. “there were lower levels of lipid peroxidation and bilirubin concentration, coupled with elevated levels of glutathione peroxidase (GPx) activity observed in mule neonates during the initial month post-foaling.“. In fact, the authors themselves later point out that GPx activity only begins to rise on day 30, while lipid peroxidation is significantly lower already after 12 hours. Still considering the fact that this is not necessarily true for both animals (we will only be able to say this when we see the graphs), this result is important and must be properly discussed, in light of the complex interactions of the different components of the antioxidant defence system. It is not sufficient to cite similar bibliographic data.

MINOR POINTS

1) Authors should use the canonical symbols for superoxide anion (•O2) and hydroxyl radical (•OH).

2) In the bibliography, Authors should make explicit all authors of each bibliographic entry, respecting the formatting required by the journal. They should also indicate DOI codes (if any).

Round 2

Reviewer 1 Report

Comments and Suggestions for Authors

Thanks to the authors for the effort in rearranging the manuscript. The authors have satisfactorily addressed almost all my concerns. I have some minor considerations (see the comments below). After these are addressed, I would recommend the manuscript for publication in Animals.

Abstract

Lines 34-35: I suggest little revision of this sentence “Parameters assessed included systemic concentrations of lipid peroxidation….” into “Parameters assessed included systemic concentrations of Thiobarbituric Acid Reactive Substances (TBARS) and carbonyl groups, activities of antioxidants: superoxide dismutase (SOD) and glutathione peroxidase (GPx), and levels of total, indirect, and direct bilirubin.”

 Line 38: part of sentence “Thiobarbituric Acid Reactive Substances (TBARS), lipid peroxidation…” change with “TBARS as marker of lipid peroxidation…”

 In Abstract add obtained result for protein oxidation and SOD.

 Materials and Methods

Lines 106-109: the explanation of the methodology for protein oxidation is redundantly here, replaced only with “carbonyl groups”.

Line 121: Add to make it clearer - ….protein peroxidation was assessed by measuring carbonyl groups level….

Line 124: replace “grouping” into “groups”.

 Results

As TBARS is a marker for lipid oxidation so are carbonyl groups are markers of protein oxidation. Please be consistent- replace protein oxidation with carbonyl groups.

Line 167: insert at the end of the sentence (data not shown).

To align with the initial part of the result, I suggest removing the subtitles: 3.1.1. Enzymatic Antioxidant Profile; 3.1.2. Superoxide Dismutase (SOD) and Glutathione Peroxidase (GPx); 3.1.3. Bilirubin and instead of that add - 3.2. Antioxidant Markers (line 168).

Tables

Line 189: add (carbonyl groups) after protein oxidation. In Table 1. replace variable - protein oxidation with carbonyl groups.

Author Response

Dear Reviewer,

Thank you for reviewing our manuscript and your valuable feedback. We have answered and made changes per your recommendations and feedback. Please find our responses in red.

Abstract

Lines 34-35: I suggest little revision of this sentence “Parameters assessed included systemic concentrations of lipid peroxidation….” into “Parameters assessed included systemic concentrations of Thiobarbituric Acid Reactive Substances (TBARS) and carbonyl groups, activities of antioxidants: superoxide dismutase (SOD) and glutathione peroxidase (GPx), and levels of total, indirect, and direct bilirubin.” 

 Line 38: part of sentence “Thiobarbituric Acid Reactive Substances (TBARS), lipid peroxidation…” change with “TBARS as marker of lipid peroxidation…”

 In Abstract add obtained result for protein oxidation and SOD.

A: All modifications/inclusions are made

 Materials and Methods

Lines 106-109: the explanation of the methodology for protein oxidation is redundantly here, replaced only with “carbonyl groups”.

Line 121: Add to make it clearer - ….protein peroxidation was assessed by measuring carbonyl groups level….

Line 124: replace “grouping” into “groups”.

A: All modifications/inclusions are made

 Results

As TBARS is a marker for lipid oxidation so are carbonyl groups are markers of protein oxidation. Please be consistent- replace protein oxidation with carbonyl groups. 

Line 167: insert at the end of the sentence (data not shown).

To align with the initial part of the result, I suggest removing the subtitles: 3.1.1. Enzymatic Antioxidant Profile; 3.1.2. Superoxide Dismutase (SOD) and Glutathione Peroxidase (GPx); 3.1.3. Bilirubin and instead of that add - 3.2. Antioxidant Markers (line 168).

Tables

Line 189: add (carbonyl groups) after protein oxidation. In Table 1. replace variable - protein oxidation with carbonyl groups. 

A: All modifications/inclusions are made

Reviewer 3 Report

Comments and Suggestions for Authors

The Authors took into account are part of my suggestions. Below I keep the numerical indication of the previous ones.

MAJOR POINTS

2) Sorry to insist, but what does it mean that there is no interaction between time and group for any of the variables? That there is no significant linear correlation? That was already evident from the graphs, just as it is very evident from the table that there is a significant difference between mule and horse at 30 days for some paramenters, and that it is possible that there is for other times as well. Therefore, it is imperative that the authors present separate graphs for the two organisms and if, as I think, there are other statistically significant differences (between groups or between times for the same group), that this be discussed thoroughly.

3) “We agree that this is an important part of the discussion and have that information on Line 311-312.” The line numbers quoted do not correspond to a part in the main text of the new version of the manuscript, but I speculate that the Authors are referring to the following text: “which can be related to a different pattern of peroxidation in the different species.”. This is a completely generic sentence and before any specific hypothesis supported by bibliography. I renew my request for a proper discussion of this result in light of the complex interactions of the different components of the antioxidant defense system.

MINOR POINTS

1) Probably the text I sent has been modified by the system. The canonical symbols for superoxide anion and hydroxyl radical are •O2 and •OH, respectively.

Author Response

Dear Reviewer,

Thank you for your valuable feedback. We have addressed your suggestions and responses can be found below in blue.

MAJOR POINTS

2) Sorry to insist, but what does it mean that there is no interaction between time and group for any of the variables? That there is no significant linear correlation? That was already evident from the graphs, just as it is very evident from the table that there is a significant difference between mule and horse at 30 days for some parameters, and that it is possible that there is for other times as well. Therefore, it is imperative that the authors present separate graphs for the two organisms and if, as I think, there are other statistically significant differences (between groups or between times for the same group), that this be discussed thoroughly.

A: Dear reviewer as we described in statistical methodology, we analyzed the variables using a linear mixed model for repeated measures over time (PROC MIXED), with group and age as fixed effects. In this case, the absence of interaction indicates that both species behaved similarly throughout the moments of evaluation. In this case, when evaluating the effect of time, both species should be analyzed as a single group. In this case, a difference that occur within moments of evaluation will be verified similarly in both species. However, we agree that this point was not clear in the manuscript. Also, a point that may have confused the reviewer is that we mistakenly omitted the p value for time in the interaction table. Therefore, we corrected the text which was highlighted in this manuscript version (yellow).

3) “We agree that this is an important part of the discussion and have that information on Line 311-312.” The line numbers quoted do not correspond to a part in the main text of the new version of the manuscript, but I speculate that the Authors are referring to the following text: “which can be related to a different pattern of peroxidation in the different species.”. This is a completely generic sentence and before any specific hypothesis supported by bibliography. I renew my request for a proper discussion of this result in light of the complex interactions of the different components of the antioxidant defense system.

 A: We included a new explanation for the data, and we totally agree that was generic, sorry about it.

MINOR POINTS

1) Probably the text I sent has been modified by the system. The canonical symbols for superoxide anion and hydroxyl radical are •O2 and •OH, respectively.

A: All modifications are made